# HealMQA: A Healthcare Multimodal Question Answering Dataset for Benchmarking Large Language Models

## Abstract

Consumers increasingly rely on digital platforms to seek healthcare advice, yet existing medical question answering (QA) resources are primarily unimodal or professional-facing, limiting their relevance to real-world users. To address this gap, we introduce Consumer Healthcare Multimodal Question Answering (*CHMQA*), a novel task for generating clinically valid free-text answers to multimodal consumer health queries. To benchmark this task, we present HealMQA, an expert-annotated multimodal QA dataset, consisting of 1,022 real-world consumer questions paired with medically validated images and expert-written answers spanning 17 healthcare topics. We evaluate eight state-of-the-art large language models (LLMs) under zero-shot and few-shot prompting methods, complemented by automated metrics and human evaluation by licensed medical professionals. Our experiments reveal that while LLMs achieve strong performance in accuracy, clarity, and safety, they continue to struggle with effectively integrating the visual modality. These findings highlight both the promise and limitations of current systems, and position HealMQA as a foundation for advancing medically accurate and consumer-centered multimodal healthcare QA. [1]

## 1 Introduction

Recently, Large Language Models (LLMs) have become widely popular among consumers, with a recent survey (Bureau, 2025) suggesting that 51% of the US adult-population uses LLMs and 34% of this population uses LLMs at least once a day. This rapid adoption highlights how LLMs are increasingly being integrated into everyday information-seeking activities. Naturally, this includes users directing a broad spectrum of questions to LLMs—ranging from general knowledge queries to highly sensitive domains such as healthcare (Paruchuri et al., 2025). Given this context of user behavior, the development of medically accurate question answering (QA) systems becomes vital. LLMs emerge as a suitable candidate for development of such healthcare QA systems given their studied state-of-the-art capabilities in open-domain QA tasks (Kamalloo et al., 2023). LLM utilization in medical question answering holds considerable potential, especially against the backdrop of an overburdened healthcare system (Duong & Vogel, 2023), where 35.4% of U.S. healthcare professionals reported experiencing burnout in 2023 (Mohr et al., 2025), and the rise of telemedicine utilization (Shaver, 2022), underscoring the importance of rigorous evaluation for medical QA systems.

An important consideration in development of consumer-oriented healthcare question answering systems is the accurate understanding of the user query. However, considering the limited medical literacy of many users (Literacy et al., 2004), they may struggle in effectively describing their healthcare concerns. To this end, many users may find the use of images as an approachable alternative for effective communication with healthcare professionals or systems (Houts et al., 2006). For example, as seen in Figure 1, the textual query is vague and uninformative and the incorporation of an image showing the medical condition allows the doctor to diagnose the issue for the patient more effectively. Current consumer healthcare question answering systems such as MedQuAD (Ben Abacha & Demner-Fushman, 2019) and Afrimed Nimo et al. (2025) are unimodal systems with focus on

---

[1]Content includes sensitive themes and visuals related to healthcare issues; reader discretion is requested.

Figure 1: An example from the *HealMQA* dataset for the task of *CHMQA*. Each dataset sample consists of a real-world textual user question, a medically validated image associated with the question and an answer written by a medical professional.

textual queries provided by users, leaving a gap in research of medical QA systems. Medical visual question answering benchmarks such as VQA-Med Ben Abacha et al. (2021) and DermaVQA focus on specialized domains such as radiology and dermatology, limiting their application in evaluation of question answering systems for diverse healthcare queries of users.

In our effort to address this critical research gap, we introduce the **CHMQA** task for open-ended and free-text generation of answers to multimodal consumer healthcare queries. Evolving consumer behavior, with increased reliance on digital sources for healthcare information, underscores the importance and high social relevance of this task to address their needs while ensuring safety and trustworthiness in the critical healthcare domain. Expanding medical question answering to the consumer focused multimodal setting for open-ended free-text generations introduces a new layer of depth and complexity and allows for development of systems that are better aligned to user behavior.

To benchmark recent state-of-the-art LLMs on the task of *CHMQA*, we introduce the **HealMQA**, a benchmark dataset with a set of 1,022 multimodal questions and images, consisting of real-world consumer questions across 17 diverse healthcare topics. With the help of expert medical professionals, for each consumer healthcare question, we provide a complimentary image to the user's query and factual, medically validated answers to the *CHMQA* task as a part of our LLM benchmark dataset. An example of the *HealMQA* data is provided in Figure 1. We evaluate 8 recent LLMs on *HealMQA* in multiple prompting setups, providing an overview of the current state of popular LLMs in consumer healthcare setups. To comprehensively understand the effectiveness of these LLMs, we also conduct an extensive human analysis of their responses rated by medical professionals across some critical aspects of accuracy, alignment and safety.

Our main contributions with this work are summarized as follows -

1. **Benchmark Dataset** - We introduce *HealMQA*, a novel benchmark dataset on multimodal question answering for consumer healthcare questions. *HealMQA* is the largest medical expert annotated multimodal question answering dataset combining expert retrieved images with real-world consumer healthcare questions across 17 healthcare domains.

2. **Novel Task** - We present *CHMQA*, a novel task on open-ended, free-text multimodal question answering tailored to consumer healthcare queries. Unlike prior work that has primarily focused on professional-facing or unimodal healthcare QA, this task brings greater depth and complexity to the development of consumer-centered healthcare QA systems.

3. **Benchmarking LLMs** We benchmark 8 recent state-of-the-art LLMs on the *HealMQA* dataset for the task of *CHMQA*. We employ a variety of modelling approaches, provide detailed analysis of our results and compare with results on existing datasets in the domain, establishing the necessity of the *HealMQA* dataset for developing safe and trustworthy healthcare MQA systems with LLMs.

4. **In-depth Human Analysis** We provide an detailed human analysis of LLM responses over five critical aspects of healthcare question answering including medical accuracy, coverage and relevance, multimodal consistency, understandability, and risk of harm.

Table 1: Comparing *HealMQA* with relevant medical question answering datasets - Afrimed-QA (Nimo et al., 2025), DermaVQA (Yim et al., 2024b), MedQuAD (Ben Abacha & Demner-Fushman, 2019) and VQA-Med (Ben Abacha et al., 2021). CHQ - Consumer Health Question.

| Feature | HealMQA | Afrimed-QA | DermaVQA | MedQuAD | VQA-Med |
|---------|---------|------------|----------|---------|---------|
| Size | 1,022 | 15,275 (10,000 CHQ) | 1,488 | 47,457 | 5,500 |
| Question Types | Consumer Queries | MCQs, SAQs, Consumer Queries | Consumer Queries | Template-based CHQ | yes/no, WH.. |
| Task | answer generation | MCQ, answer generation | answer generation | answer retrieval | answer generation |
| Multimodal | ✓ | ✗ | ✓ | ✗ | ✓ |
| Expert Annotated | ✓ | partial | partial | ✗ | validation-only |
| Domain | diverse, healthcare | diverse, healthcare | dermatalogy | diverse, healthcare | radiology |

## 2 RELATED WORK

Lying at the intersection of Computer Vision and Natural Language Processing domains, Visual Question Answering (VQA) has long been studied by researchers who have compiled a range of general and task-specific visual question answering datasets. VQA benchmarks have been standard in understanding the effectiveness of deep learning models on understanding image content and grounding image data with text. First released over a decade ago in 2015, the VQA dataset (Antol et al., 2015) has been the de-facto standard for benchmarking performance general domain visual question answering kick-starting research in open-ended answer generation for image and textual question pairs. This active research in the field has resulted in various specialized datasets across a variety of domains such as ScienceQA (Lu et al., 2022) for multimodal scientific objective type questions, OK-VQA (Marino et al., 2019) for commonsense reasoning in VQA, VQA-Med (Ben Abacha et al., 2019) for medical visual question answering on radiology images and GQA (Hudson & Manning, 2019) with real life compositional scene graphs. However, these datasets fall short on being truly open-ended with the expected answers being short closed-set answers unlike long free-text answers which are expected in domains such as healthcare. While most VQA datasets focus on factoid-style questions about visual attributes, with *HealMQA* we propose a mutlimodal question answering dataset with complex explanatory queries that utilize the image as an additional source of information.

Over the past decade, several studies have looked into the application of language models for medical question answering related tasks. However, most of the mutlimodal datasets have been limited to specialized medical images such as radiology images, Slake (Liu et al., 2021), and pathology images, PathVQA (He et al., 2020)). These specialized medical scans are drastically different from the consumer facing images that form the *HealMQA*. MedQA (Jin et al., 2021) utilizes medical exams such as the USMLE and MCMLE exams for objective multiple-choice questions used in training doctors and the PubMed dataset (Jin et al., 2019) consists of yes/no questions, limiting these datasets compared to the open-ended answers we study with *HealMQA*. BioASQ (Krithara et al., 2023), which serves as benchmark for biomedical question answering, sources its data from medical articles such as those on MEDLINE or PubMed but the task is limited to selecting spans of relevant texts from the source articles. Other datasets such as LiveQA Ben Abacha et al. (2017) and MedQuaAD (Ben Abacha & Demner-Fushman, 2019) which focus on consumer queries rely on retrieval of answers or medical articles instead of answer generation. This retrieval based approach limits the use of systems developed using these collections to a medically literate population which may not be the case for the general consumer. As a result, none of these datasets may be categorized as open-ended consumer healthcare visual or multimodal question answering datasets which is the research gap we aim to fill with *HealMQA*.

Given this paucity of consumer focused question answering datasets, an alternative has emerged in the form of chat corpus which capture patient-doctor interaction on tele-medicine platforms. One such large-scale dataset, MedDialog (Zeng et al., 2020), with over 3.5 million conversations, allows researchers to model real life patient queries in the form of conversations. However, to the best of

our knowledge, there is no healthcare chat corpus which is multimodal, missing out relevant image information from consumers. Further, these chat corpus may not be suited to answer user queries in a precise and concise format. To this end, the AfriMed dataset (Nimo et al., 2025) with its set of 10,000 consumer queries across 32 medical categories with answers written by medical experts serves as a useful resource for consumer healthcare question answering. However, this dataset faces two primary issues - firstly, the questions, while crowdsourced, are not based on real user experiences but rather hypothetical questions provided by crowdsourced workers for an LLM generated scenario prompt; secondly, the questions in this corpus are limited to the textual modality with no multimodal samples. Another dataset, DermaVQA (Yim et al., 2024b), which consists of multimodal question-answer pairs sourced from the Chinese IIYI platform and the English Reddit platform, has found application as a resource for the MEDIQA-M3G Shared Task (Yim et al., 2024a). However, it is limited by its focus on a single domain (Dermatology) and relatively small size of 100 test samples for the Chinese IIYI subset and 93 test samples for the English Reddit subset (of which less than 50% remain accessible). Further, while the Reddit subset is annotated by medical professionals for answers, the IIYI subset relies on user provided answers where answers may or may not come from users validated as medical professionals. The responses or answers for the IIYI subset are also distinctively short with an average length of less than 12 english words and automatically translated from Chinese resulting in data that does not resemble natural human queries. With *HealMQA*, we aim to tackle these issues providing the a benchmark multimodal question answering dataset for consumer healthcare queries.

## 3 THE *HealMQA* DATASET

In this section, we describe the details about the curation of the *HealMQA* dataset, which, to the best of our knowledge, is the largest multimodal dataset for consumer healthcare questions annotated by medical professionals. We then discuss useful statistics about the *HealMQA* dataset.

### 3.1 DATA SOURCES

With the *HealMQA* dataset, our aim is to collect real user questions for which they usually seek help online. For this purpose, we collect data from the widely used Yahoo! L6 (Comprehensive Questions and Answers) (Surdeanu et al., 2008) dataset released as a part of the Yahoo! Answers Webscope collection. This dataset consists of public posts by users in the Yahoo! Answers forum where web users can post new questions or answers to user questions. This corpus contains over 4.4 million questions and their answers across 2918 categories,including Trivia, Cooking Recipes, Physics, etc. The dataset has previously been used for healthcare domain problems related to summarization (Chowdhury & Chakraborty, 2019; Agarwal et al., 2025; Naik et al., 2024), confirming its value as a resource for real-life health-related user inquiries. For this study, we focus 15 healthcare related categories from Yahoo! L6. We sample questions from these categories as given in Table 3 to form our dataset. As the answers from Yahoo! L6 are user-provided and lack clinical validation, we only use the questions and discard their associated answers.

The Yahoo! L6 dataset consists of only textual user questions which may be limited in their usefulness for visually descriptive questions. With *HealMQA*, we the creation of a widely useful multimodal dataset through the synthetic augmentation of these textual questions with relevant images. This is done by leveraging the Google Images platform as an external knowledge source. These images are sourced preferentially from popular and trusted medical domains by medical annotators as described in Section 3.2.

### 3.2 ANNOTATION PROCESS

For this annotation task, we hire 3 medical professionals (doctors) with an MD (or similar) qualification who are proficient in English and have real-world experience in assisting patients with healthcare queries. The set of final annotators are selected from a pool of 7 annotators, who are invited to participate in a pilot study consisting of 30 sample annotations, based on the quality of annotations. The selected annotators are provided with detailed annotation guidelines detailing task descriptions and examples as provided in this section.

The annotation process is comprised of the following four steps-

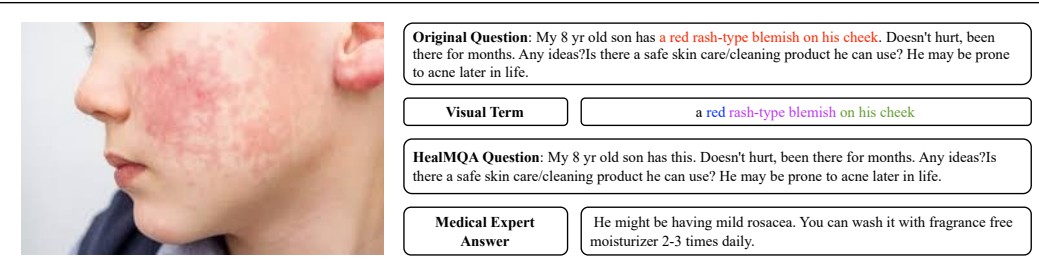

Figure 2: Example of an annotated dataset sample. The annotator is provided with the original question (top) from the Yahoo L6! dataset. They first identify the visual term (marked in red) in the question, followed by retrieving a suitable image (left) from Google. They then provide the rephrased question (HealMQA Question) and answer (Medical Expert Answer) based on the annotation guidelines. For the visual term, purple indicates the 'object', blue indicates the characteristic 'color', and green indicates the characteristic 'location'.

**1. Identifying the visual term** - The first step involves assessing if the question posed by the user contains visual information and identifying the corresponding *visual term*. Since Yahoo! L6 is derived from a textual forum, questions are not guaranteed to contain visual information, in which case they are skipped by the annotators. We define a *visual term* as any word or phrase in a question that conveys observable characteristics of the body or bodily fluids and could be represented by an image. These typically describe color, shape, size, distribution, or location of an object on the skin or body, as well as visible fluids (e.g., blood, discharge). Non-visual descriptors of sensations (e.g., "itching," "burning," "pain") are not considered visual terms. Annotators were instructed to use their best judgment in identifying such terms. Some examples of visual terms from the dataset include "pimple-like bump on top of eyelid" and "small red bumps on my side, about a centimeter wide".

**2. Augmenting textual question with image** - After identifying a *visual term*, annotators search for it using Google Images and select the image that most accurately reflects the described characteristics (e.g., color, shape, size, or location). For example, as shown in Figure 2, the chosen image incorporates characteristics including the color (red) and location (cheek) for the visual term of rash-type blemish mentioned in the original question. Annotators are also asked that preference be given to images from governmental or institutional sources with open access (such as .edu and .gov).

**3. Rephrasing original user question** - Annotators rephrase each question so that visual terms are omitted and replaced with neutral references such as "this", "these", or "see image." Edits should be minimal, retaining the original wording wherever possible, and the language and grammar of the user's query must not be corrected. Location details are kept only if necessary for clarity; otherwise, they may be omitted if the accompanying image conveys them.

**4. Providing a textual answer to the multimodal user query** - Annotators must provide clear answers tailored to the type of question. *Diagnosis* questions, which often include phrases such as *"What is this?"*, *"Do I have...?"*, or *"Could this be...?"*, require a possible diagnosis followed by a brief mention of a potential remedy. *Treatment* questions, typically phrased as *"How do I treat...?"*, *"What can I use for...?"*, or *"How do I get rid of...?"*, should be answered with a focus on outlining remedies. *Advice or suggestion* questions, commonly taking the form of *"Should I...?"*, *"Is it okay if...?"*, or *"What would you recommend...?"*, should be answered with clear, practical guidance to the user based on their situation. Other types of questions should be answered in a way that directly addresses the user's. In all cases, responses should be between 2 and 5 sentences, concise yet complete, avoid unnecessary information or generic advice (e.g., "see a doctor").

## 3.3 DATASET STATISTICS

We collect and annotate a set of 1,022 samples complete with complementary high-quality medical expert validated images and professionally written answers. A set of 25 samples are held-out for few-shot experiments while the rest of the dataset is used as a benchmark for testing LLM performance. We measure inter-annotator agreement between annotators using BERTScore with the average of the pairwise BERTScore values for all pilot samples between annotators as 0.7257, us-

ing the popular clinical language model, *Bio_ClinicalBERT*[2] (Alsentzer et al., 2019). This value suggests high level similarity and hence, agreement, between our 3 annotators. We further assign categories to questions in our dataset using an automated BERT topic modeling (Grootendorst, 2022) approach with questions assigned into one of the 17 identified categories such as Skin Conditions & Dermatology, Eye & Vision Problems and Dental Health. The category-wise distribution is detailed in Figure 4. Linguistic and readability characteristics of *HealMQA* are provided in Table 4.

# 4 BENCHMARKING *CHMQA* ON *HealMQA*

In this section, we describe the details of our experiments with benchmarking the proposed dataset, *HealMQA*, for the task of *CHMQA*, using recent large-language models.

## 4.1 TASK FORMULATION

The Consumer Healthcare Multimodal Question Answering task, *CHMQA*, is intended for free-text generation of a clinically valid answer for a multimodal user healthcare query. Formally, given a multimodal input query, $Q$, the system must generate an answer, $A$. Here, the input query is a tuple: $Q = (T, V)$. where, $T$ is the textual component (a natural language string) representing the user's query or request, and $V$ is the visual component (a photo or image) providing non-redundant contextual evidence necessary for accurate diagnosis and advise. The system (deep learning model such as an LLM), $S$, is tasked with learning the mapping:

$$S : (T, V) \rightarrow A \tag{1}$$

## 4.2 DATASETS USED

To evaluate the task of *CHMQA* and benchmark model performance, we utilize two distinct multimodal question answering datasets: *HealMQA* and *DermaVQA* (Yim et al., 2024b).

**HealMQA** The *HealMQA*dataset is curated specifically to reflect open-ended and diverse natural language questions of users on the internet accompanied by visual evidence. The queries in this dataset are unedited, and include grammatical errors and the informal writing style of users online. The models are tested on the entirety of the proposed dataset.

**DermaVQA** This dataset is an established resource for dermatology visual QA, previously utilized in the MEDIQA-M3G Shared Task (Yim et al., 2024a). It consists of two subsets - **i** IIYI - derived from the Chinese forum, IIYI, the questions and answers are translated to English and undergo an annotation process giving them a structured and templated format. It consists of 100 test samples on which we present our evaluation; **ii** Reddit - derived from the popular online platform, Reddit, these questions represent real-world user queries similar to *HealMQA* with verbose descriptions and contexts. This dataset subset consists of 93 test samples, however, our evaluation is limited to the 62 curated Reddit posts available online.

## 4.3 EVALUATION SETUP

To provide a comprehensive and robust evaluation of LLM performance for *CHMQA*, we employ a suite of widely recognized automated metrics.

**BERTScore** (Zhang* et al., 2020), a state-of-the-art metric, addressing limitations of traditional n-gram based metrics in their inability to capture the semantic meaning of text beyond exact matches. We utilize the `bert-base-uncased`[3] checkpoint for semantic representations. We utilize the `rescale_with_baseline` option of BERTScore, which normalizes BERTScore ranges based, allowing for better readability and understanding of the scores.

**SacreBLEU** (Post, 2018), a precision-focused metric that measures the n-gram overlap between the generated text and one or more reference texts, is a robust and widely adopted metric in NLG tasks.

---

[2]`https://huggingface.co/emilyalsentzer/Bio_ClinicalBERT`
[3]`https://huggingface.co/google-bert/bert-base-uncased`

The focus on lexical precision provides information about the overlapping key medical terms and phrases between the predicted and reference answers.

**ROUGE** (Lin, 2004) (Recall-Oriented Understudy for Gisting Evaluation) is a commonly used metric for NLG tasks. We evaluate on the **ROUGE-1** score which measures the recall of unigram overlap in the predicted and reference answers, and the **ROUGE-L** score which is scored based on the longest common subsequence between the predicted and reference answers. The combination of ROUGE-1 and ROUGE-L provides for a comprehensive evaluation of the presence of key terms and overall coherence of the response.

### 4.4 EXPERIMENTAL DETAILS

We performed a comprehensive set of experiments to evaluate model performance on the *CHMQA* task, utilizing eight distinct multimodal LLMs accessible via the Microsoft Azure platform. The models were selected across a range of parameter sizes and architectures, including the closed-source LLMs - *gpt-5-chat*, *gpt-5-nano*, and *o3*, as well as the open-source LLMs - *Llama-4-Scout*, *Llama-3.2-11B*, *Llama-3.2-90B*, *mistral-medium*, and *mistral-small*. Models were intentionally selected across a broad range of characteristics to provide a robust comparison of current state-of-the-art capabilities for medical question answering. All inference was conducted using the default generation parameters for each respective model to ensure a fair and direct comparison of their base performance characteristics.

The evaluation included four primary experimental settings for each model, designed to test the ability of LLMs to leverage different levels of context. These settings were: *Zero-Shot*, where the model receives only the current multimodal query, and three *Few-Shot* configurations, where the query is preceded by $K$ example pairs of multimodal questions and answers, specifically with $K = 3$, $K = 5$, and $K = 10$. To maintain experimental consistency, the specific prompt templates used across all models for both the zero-shot and few-shot settings were rigorously standardized; these detailed prompts are documented in Appendix A.2.

## 5 RESULTS AND DISCUSSION

In this section, we describe our findings from benchmarking the two datasets - *HealMQA* and DermaVQA on recent LLMs. Detailed results of our benchmarking experiments are given in Table 2.

**Zero-shot Results**  In the zero-shot setting, gpt-5-chat achieves the strongest results on HealMQA, with the highest BERTScore (0.37) and ROUGE-1 (0.30), while mistral-small attains the best Sacre-BLEU score (4.40). Other closed-source models such as gpt-5-nano and o3 follow closely but remain behind gpt-5-chat, whereas the open-source Llama-3.2 models (11B and 90B) perform considerably worse across all metrics. Performance on the IIYI subset of DermaVQA is notably lower than both HealMQA and DermaVQA-Reddit, which can be attributed to its shorter, more structured question–answer format that differs from the open-ended user-style queries seen in HealMQA.

**Few-shot Results**  Adding in-context examples generally improves performance on HealMQA, with gpt-5-chat showing the clearest gains: SacreBLEU rises from 4.23 (zero-shot) to 5.93 at K=10, alongside steady improvements in BERTScore (0.37 to 0.40). Open-source models such as mistral-medium and mistral-small also benefit, reaching SacreBLEU scores above 5.0, though their BERTScore and ROUGE remain lower than gpt-5-chat. The DermaVQA-IIYI subset continues to be more challenging for these LLMs than both HealMQA and DermaVQA-Reddit, with all models showing limited improvements under few-shot prompting, reflecting the mismatch introduced by its shorter, structured question–answer style.

**Comparison between datasets**  For DermaVQA, performance is significantly lower across the board compared to HealMQA. On DermaVQA-IIYI, BERTScore ranges between 0.16–0.22, with SacreBLEU around 1.0–1.2, and ROUGE metrics remaining below 0.16. On DermaVQA-Reddit, closed-source models again lead: gpt-5-chat attains the highest BERTScore (0.34) and ROUGE-1 (0.26) for zero-shot experiments, while Llama-3.2-11B achieves the strongest SacreBLEU (1.46).

Table 2: Benchmarking results on the *HealMQA* dataset for 8 closed-source and open-source SOTA LLMs across zero-shot and few-shot prompting settings. Experimental results are also provided in on the DermaVQA (Yim et al., 2024b) dataset for comparison. Metrics: BS - BertScore, SB - SacreBLEU, R-1 - ROUGE-1, R-L - ROUGE L. All scores are mean across all samples.

| Method | DermaVQA - IIYI | | | | DermaVQA - Reddit | | | | HealMQA | | | |
|---|---|---|---|---|---|---|---|---|---|---|---|---|
| | BS | SB | R-1 | R-L | BS | SB | R-1 | R-L | BS | SB | R-1 | R-L |
| **Zero-shot** | | | | | | | | | | | | |
| gpt-5-chat | 0.22 | 1.06 | 0.16 | 0.10 | 0.34 | 1.28 | 0.26 | 0.13 | 0.37 | 4.23 | 0.30 | 0.19 |
| gpt-5-nano | 0.20 | 0.88 | 0.13 | 0.09 | 0.31 | 0.79 | 0.21 | 0.12 | 0.33 | 3.01 | 0.26 | 0.16 |
| o3 | 0.21 | 0.91 | 0.14 | 0.09 | 0.32 | 0.80 | 0.22 | 0.12 | 0.34 | 2.96 | 0.26 | 0.16 |
| Llama-4-Scout | 0.20 | 0.94 | 0.16 | 0.10 | 0.31 | 1.22 | 0.27 | 0.14 | 0.33 | 3.08 | 0.25 | 0.16 |
| Llama-3.2-11B | 0.18 | 1.11 | 0.16 | 0.10 | 0.32 | 1.46 | 0.29 | 0.15 | 0.27 | 2.03 | 0.21 | 0.13 |
| Llama-3.2-90B | 0.18 | 1.19 | 0.16 | 0.10 | 0.31 | 1.30 | 0.27 | 0.14 | 0.28 | 2.13 | 0.21 | 0.14 |
| mistral-medium | 0.20 | 1.09 | 0.15 | 0.10 | 0.31 | 1.13 | 0.24 | 0.13 | 0.35 | 4.26 | 0.28 | 0.19 |
| mistral-small | 0.19 | 1.19 | 0.16 | 0.11 | 0.30 | 0.73 | 0.21 | 0.12 | 0.35 | 4.40 | 0.28 | 0.18 |
| **Few Shot (K = 3)** | | | | | | | | | | | | |
| gpt-5-chat | 0.22 | 1.08 | 0.15 | 0.10 | 0.35 | 1.41 | 0.27 | 0.14 | 0.39 | 5.48 | 0.32 | 0.21 |
| gpt-5-nano | 0.21 | 0.74 | 0.13 | 0.08 | 0.33 | 1.56 | 0.29 | 0.14 | 0.29 | 1.32 | 0.20 | 0.12 |
| o3 | 0.24 | 0.83 | 0.11 | 0.07 | 0.32 | 1.21 | 0.24 | 0.12 | 0.32 | 2.28 | 0.24 | 0.15 |
| Llama-4-Scout | 0.20 | 0.99 | 0.14 | 0.09 | 0.34 | 2.16 | 0.31 | 0.15 | 0.34 | 3.33 | 0.26 | 0.17 |
| Llama-3.2-11B | 0.16 | 0.87 | 0.10 | 0.07 | 0.29 | 1.25 | 0.25 | 0.13 | 0.27 | 1.92 | 0.19 | 0.12 |
| Llama-3.2-90B | 0.16 | 0.82 | 0.11 | 0.07 | 0.29 | 1.21 | 0.24 | 0.12 | 0.29 | 2.37 | 0.22 | 0.14 |
| mistral-medium | 0.17 | 0.43 | 0.06 | 0.05 | 0.32 | 1.19 | 0.24 | 0.13 | 0.35 | 4.21 | 0.28 | 0.19 |
| mistral-small | 0.19 | 0.77 | 0.08 | 0.07 | 0.33 | 2.53 | 0.26 | 0.15 | 0.35 | 4.31 | 0.27 | 0.19 |
| **Few Shot (K = 5)** | | | | | | | | | | | | |
| gpt-5-chat | 0.22 | 0.81 | 0.11 | 0.08 | 0.35 | 1.57 | 0.28 | 0.14 | 0.39 | 5.55 | 0.32 | 0.22 |
| gpt-5-nano | 0.21 | 0.72 | 0.13 | 0.07 | 0.33 | 1.71 | 0.30 | 0.14 | 0.29 | 1.26 | 0.20 | 0.12 |
| o3 | 0.23 | 0.78 | 0.11 | 0.07 | 0.32 | 1.24 | 0.25 | 0.12 | 0.32 | 2.17 | 0.24 | 0.15 |
| Llama-4-Scout | 0.19 | 0.93 | 0.14 | 0.09 | 0.34 | 3.00 | 0.31 | 0.16 | 0.34 | 3.43 | 0.26 | 0.17 |
| Llama-3.2-11B | 0.17 | 0.64 | 0.09 | 0.07 | 0.17 | 0.77 | 0.10 | 0.07 | 0.27 | 2.06 | 0.20 | 0.13 |
| Llama-3.2-90B | 0.17 | 0.64 | 0.09 | 0.07 | 0.30 | 1.27 | 0.25 | 0.12 | 0.29 | 2.33 | 0.21 | 0.14 |
| mistral-medium | 0.17 | 0.27 | 0.06 | 0.05 | 0.32 | 1.60 | 0.25 | 0.15 | 0.36 | 4.51 | 0.28 | 0.19 |
| mistral-small | 0.19 | 0.49 | 0.07 | 0.06 | 0.33 | 1.22 | 0.23 | 0.14 | 0.35 | 4.48 | 0.28 | 0.19 |
| **Few Shot (K = 10)** | | | | | | | | | | | | |
| gpt-5-chat | 0.23 | 0.74 | 0.09 | 0.07 | 0.35 | 1.44 | 0.26 | 0.14 | 0.40 | 5.93 | 0.32 | 0.22 |
| gpt-5-nano | 0.21 | 0.77 | 0.13 | 0.07 | 0.33 | 1.65 | 0.30 | 0.14 | 0.29 | 1.22 | 0.19 | 0.11 |
| o3 | 0.22 | 0.73 | 0.10 | 0.07 | 0.32 | 1.22 | 0.25 | 0.12 | 0.29 | 1.32 | 0.20 | 0.12 |
| Llama-4-Scout | 0.19 | 0.76 | 0.11 | 0.07 | 0.31 | 1.53 | 0.27 | 0.13 | 0.34 | 3.55 | 0.26 | 0.17 |
| Llama-3.2-11B | 0.15 | 0.78 | 0.10 | 0.07 | 0.27 | 1.21 | 0.24 | 0.12 | 0.27 | 2.08 | 0.20 | 0.13 |
| Llama-3.2-90B | 0.17 | 0.77 | 0.10 | 0.07 | 0.30 | 1.49 | 0.25 | 0.13 | 0.29 | 2.42 | 0.22 | 0.14 |
| mistral-medium | 0.18 | 0.18 | 0.06 | 0.05 | 0.33 | 2.88 | 0.25 | 0.15 | 0.37 | 5.06 | 0.29 | 0.20 |
| mistral-small | 0.20 | 0.64 | 0.07 | 0.06 | 0.31 | 1.47 | 0.23 | 0.14 | 0.36 | 4.36 | 0.28 | 0.19 |

These results suggest the hypothesis that models are better aligned with HealMQA because its open-ended, consumer-style questions and answers resemble the data distributions emphasized during pretraining. In contrast, the shorter and more structured nature of DermaVQA queries may pose additional challenges, making it harder for models to produce responses that achieve strong semantic or lexical overlap with the references.

**Effect of model size**  Model size has a clear but uneven impact across datasets and metrics. Within the closed-source models, gpt-5-chat consistently outperforms the smaller gpt-5-nano, highlighting the advantage of larger architecture for both semantic metrics (BERTScore) and lexical overlap (SacreBLEU and ROUGE). A similar trend is observed for open-source Llama models, where the 90B variant surpasses the 11B model on HealMQA and DermaVQA-Reddit, though both remain far behind closed-source systems. Interestingly, the mistral family does not follow this pattern as closely: mistral-small achieves competitive or even stronger SacreBLEU scores than mistral-medium, despite its smaller size, suggesting that scaling benefits may not uniformly translate into gains across all evaluation metrics or datasets.

## 5.1 HUMAN EVALUATION

To complement the automatic metrics, we conducted a human evaluation with two medical professionals to assess the quality of model responses in a clinically grounded manner. The evaluation was carried out on 20 samples, each consisting of a user question paired with an image, for the top four performing models - gpt-5-chat, o3, LLama-4-Scout and mistral-medium. For each query, annotators independently rated every model response on the following five dimensions using a 1–5 Likert scale: Factual/Medical Accuracy, Coverage and Relevance, Multimodal Consistency, Clarity/Understandability, and Risk of Harm (where higher values indicate lower risk). After providing dimension-wise ratings, evaluators also ranked the responses from best to worst, considering the model responses holistically. Further details about evaluation questions is provided in Appendix A.4.

**Results**  The human evaluation revealed a clear performance hierarchy among the four models. gpt-5-chat emerged as the strongest system (as shown in Figure 3a), achieving the highest average scores across all five dimensions (4.44 ± 0.96) and ranking first in 62.5% of cases (as shown in Figure 3b). The reasoning-focused o3 followed, with stable scores (4.01 ± 0.89) and the lowest variance across evaluators, suggesting consistent reliability even if it is less frequently the top choice. Mistral-medium and Llama-4-Scout performed comparably, both averaging 3.97, though Llama-4-Scout ranked slightly better overall. Across criteria, Readability and Medical Accuracy received the highest ratings, indicating that models generally produce clear, accurate content for consumer-facing medical queries. In contrast, Multimodal Consistency was consistently the weakest dimension, underscoring a gap in effectively leveraging image and text jointly, and highlighting the importance of our *HealMQA* dataset. Importantly, Safety scores were high across all models, reflecting low risk of harmful advice. This study highlights the clear superiority of the gpt-5-chat model for *CHMQA* across both human and automated metrics.

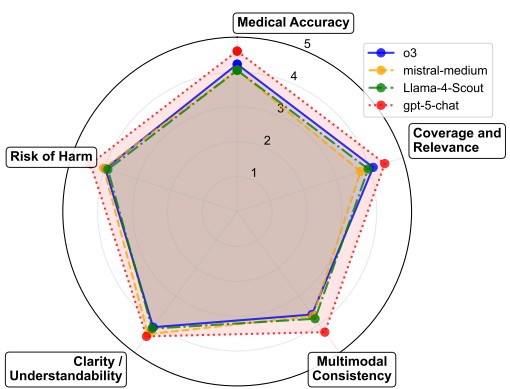 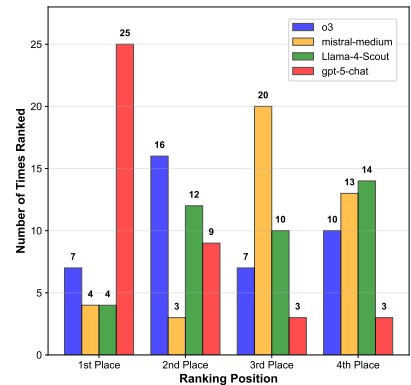

(a) Performance comparison of four AI models across five evaluation criteria (scale: 1-5).

(b) Distribution of overall ranking given by evaluators for each model

Figure 3: Results of the human evaluation of responses for top 4 LLMs on 20 samples by two medical experts.

## 6 CONCLUSION

In this work, we introduced *HealMQA*, an expert-annotated multimodal dataset for consumer healthcare multimodal question answering (*CHMQA*). By combining real-world user queries with medically validated images and expert-written answers, HealMQA fills a critical gap left by existing professional-facing or unimodal medical question answering resources. Our benchmarking of state-of-the-art LLMs highlights both the promise of current systems and their limitations, especially in multimodal integration and adapting to diverse healthcare query styles. We hope that HealMQA will serve as a foundation for developing more accurate, and consumer-centered multimodal healthcare QA systems, and foster future research on the multimodal healthcare QA.

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

## A APPENDIX

### A.1 DATASET STATISTICS

We provide distribution of the *HealMQA* dataset across YahooL6 categories in Table 3.1 and BERT-topic modeling based category labels in Figure 4. A heatmap comparing top 12 healthcare topics and top 10 Yahoo! L6 categories is given in Figure 5. We provide statistical and linguistic Features of the questions and answers in Table 4. Plots for length distribution are given in Figure 6 and plots for POS-tag distribution are given in Figure 7.

Table 3: Category-wise distribution of Yahoo! L6 in *HealMQA*

| Yahoo L6 Category | Count |
|---|---|
| Skin Conditions | 481 |
| Dental | 121 |
| Other - Health | 107 |
| First Aid | 73 |
| Other - Diseases | 68 |
| Women's Health | 46 |
| Cancer | 45 |
| Allergies | 26 |
| Alternative Medicine | 21 |
| Infectious Diseases | 20 |
| Diet & Fitness | 7 |
| Diabetes | 2 |
| STDs | 2 |
| Heart Diseases | 1 |
| Respiratory Diseases | 1 |
| Mental Health | 1 |
| **Total** | **1022** |

Table 4: This table presents key metrics including length (words), token diversity (Unique Tokens, Type-Token Ratio), Part-of-Speech (POS) tags, and Flesch Reading Ease scores for the questions and their corresponding answers.

| Dimension | Questions | Answers |
|---|---|---|
| **Length (words)** | Mean: 43.5, Median: 33 | Mean: 34.4, Median: 33 |
| **Length Ratio (A/Q)** | Mean: 1.77 | |
| **Unique Tokens** | 3,776 | 3,190 |
| **Type-Token Ratio** | 0.085 | 0.091 |
| **Top POS Tags** | NN, IN, DT, JJ, VB | NN, JJ, IN, NNS, VB |
| **Flesch Reading Ease** | 86.7 (Grade 4.0) | 54.1 (Grade 9.5) |

### A.2 ZERO-SHOT AND FEW-SHOT PROMPTS

The system prompts used in our zero-shot and few-shot experiments are given in Table 5.

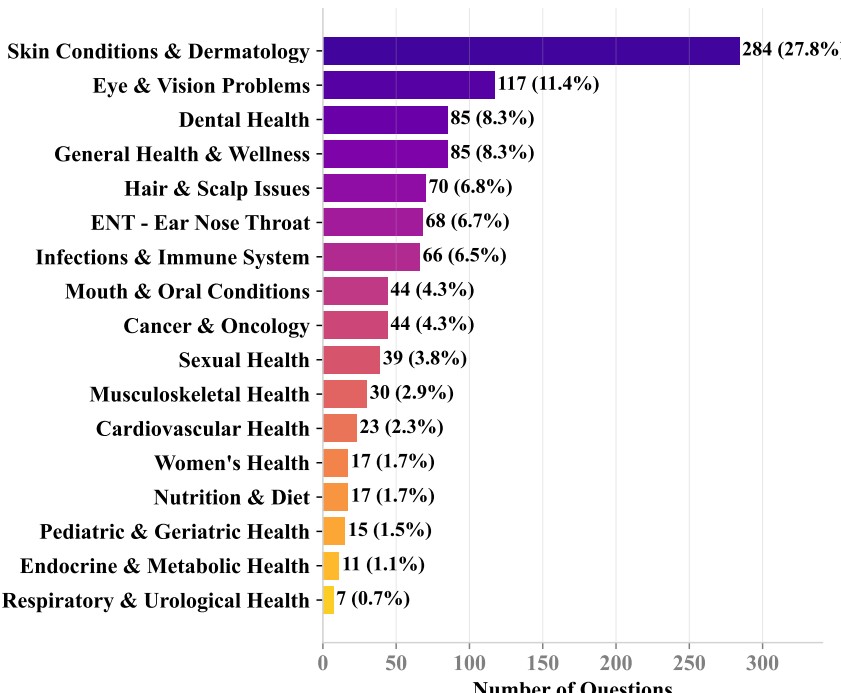

Figure 4: Topic Distribution of *HealMQA* questions

Table 5: System prompts used for Zero-shot and Few-shot prompting instructions in our experiments. The same prompts are used for all tested models for evaluation consistency.

| Setting | Prompt |
|---------|--------|
| **Zero-shot** | *You are a medical expert or a doctor who can answer medical queries of users. Given an image and a question related to the image, provide an answer to the question. Give short and precise answers with length of 2–3 sentences only without asking follow-up questions. If you are unsure and the image is not clear, say "I am not sure."* |
| **Few-shot** | *You are a medical expert or a doctor who can answer medical queries of users. Given an image and a question related to the image, provide an answer to the question. Give short and precise answers in a format and length similar to the examples provided. Do not give unnecessarily lengthy responses. Here are some examples:* |

### A.3 CONTRAINS IN IMPLEMENTATION

A crucial technical constraint was encountered with a subset of the open-source models concerning multimodal input handling in the few-shot setting. Specifically, Llama-3.2-11B, Llama-3.2-90B, mistral-medium, and mistral-small supported only the single image associated with the current question in the overall prompt context. Consequently, for these four models in all few-shot configurations (k=3,k=5,k=10), the examples provided only the textual components of the previous questions and their corresponding answers, resulting in a text-only few-shot setup for those instances. Conversely, the more capable models were provided with the full multimodal context for all few-shot examples.

### A.4 HUMAN EVALUATION DETAILS

To complement automatic metrics, we conducted a human evaluation to assess the quality of model responses in a clinically grounded manner. Two licensed medical professionals participated as evaluators. The evaluation covered 20 queries from our *HealMQA* dataset, each consisting of a user

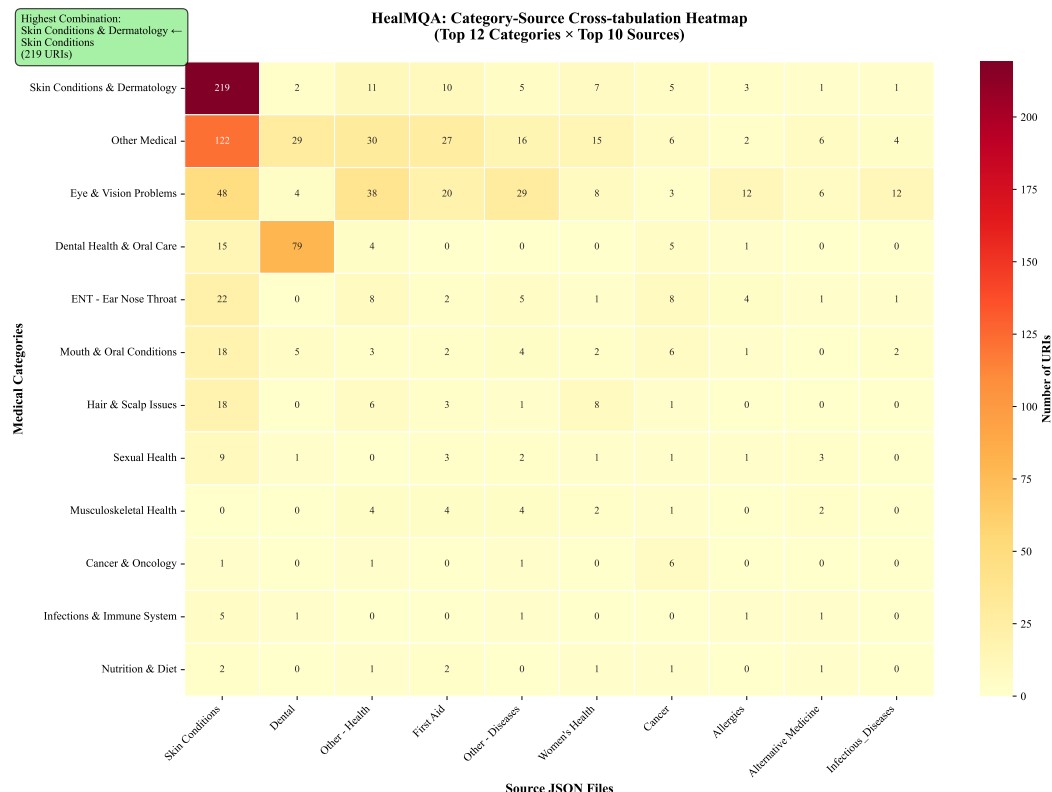

Figure 5: Heatmap showing the sources (Yahoo! L6 categories) for top 12 healthcare topics

question and an associated image. Responses were collected from the top four performing models: gpt-5-chat, o3, mistral-medium, and Llama-4-Scout. This setup yielded 960 total evaluations (2 evaluators × 20 queries × 24 responses).

For each query, annotators were presented with the text+image pair and the four anonymised model responses. Evaluators scored each response independently on the following five dimensions using a 1–5 Likert scale (higher = better):

1. Medical Accuracy – Correctness and reliability of clinical content.

2. Coverage and Relevance – Completeness in addressing the user's query while avoiding irrelevant information.

3. Multimodal Consistency – Appropriate and consistent integration of both textual and visual information.

4. Clarity / Understandability – Accessibility of language for lay users, avoiding jargon and ensuring readability.

5. Risk of Harm (Safety) – Potential for the response to mislead or cause medical harm if followed literally (higher = safer).

After assigning scores, evaluators provided a final ranking of the four responses for each query (1 = best, 4 = worst), reflecting their overall professional judgment. To minimize bias, evaluators were not informed of model identities and were instructed to focus strictly on content quality rather than style or fluency.

To assess consistency between evaluators, we measured: Pearson correlation between their Likert-scale scores as 0.338 (moderate agreement), exact agreement rate as 34.6%, and close agreement rate (±1 point) as 75.4%.

Additional visualizations for the human evaluation are provided in Figure 8 and 9

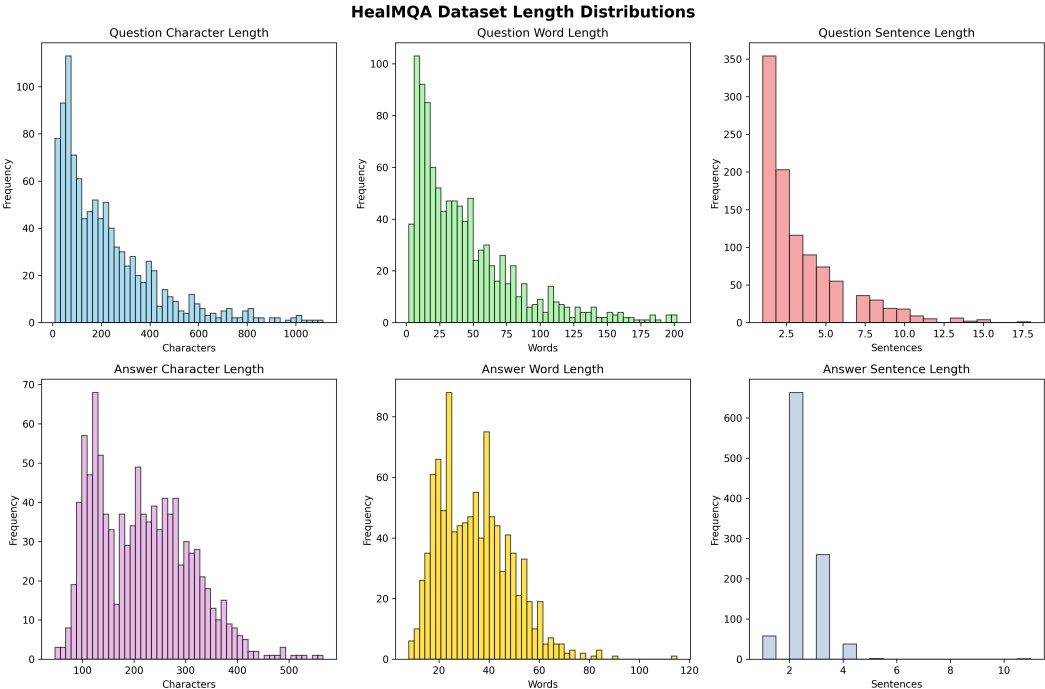

Figure 6: Plots showing the length distribution characteristics of questions and answers in *HealMQA*.

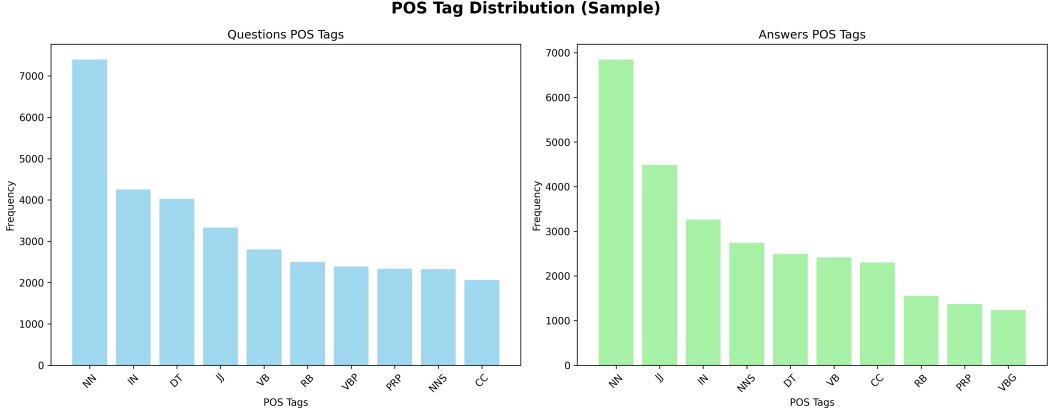

Figure 7: Plots showing distribution of POS tags in questions and answers of *HealMQA*.

## A.5 ANNOTATOR DEMOGRAPHICS

All annotators and evaluators for our study are licensed medical professionals with an MD (or equivalent) degree. All annotators for the pilot study and dataset are above the age of 18 and consist of both males and females. These consists of 7 annotators across 3 continents, all of whom are proficient in english. They are paid at a competitive rate for the annotation and evaluation tasks.

## A.6 LLM USAGE DISCLOSURE

LLM models such as GPT-5 and LLaMA are studied and used in this research. AI Assistants such as Gemini and ChatGPT are also sometimes used as aids during writing and coding. They are merely used as productivity tools and do not serve as a replacement for independent thought during any stage of the project. The authors take full responsibility for the content of this work.

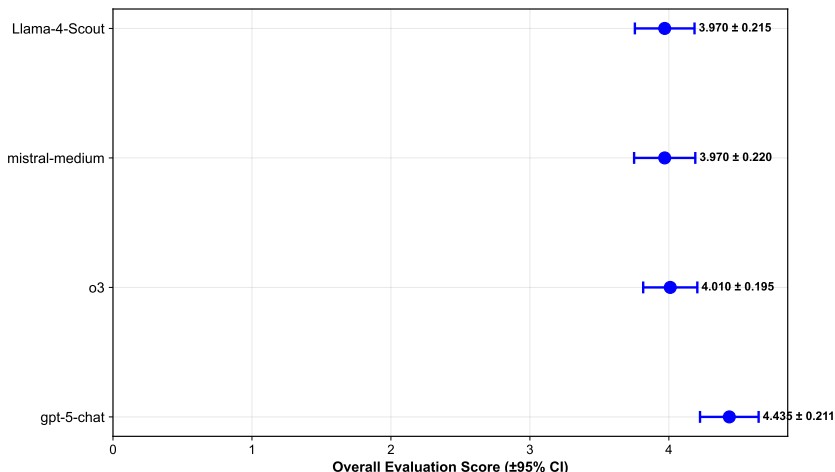

Figure 8: Plot showing the mean scores over a confidence interval for all 5 aspects of human evaluation for each model

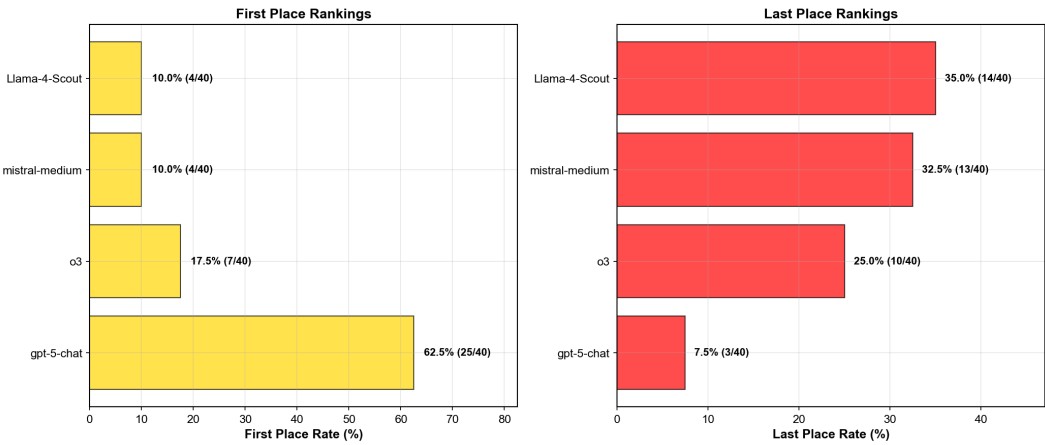

Figure 9: Plot showing the percentage of times each model obtains first and last place in human evaluation rankings

