# OpenReview forum: "HealMQA: A Healthcare Multimodal Question Answering Dataset for Benchmarking Large Language Models"
_ICLR.cc/2026/Conference — Submitted to ICLR 2026_

### Official Review · Reviewer_ws9F · 2025-10-23

**Soundness:** 2
**Presentation:** 3
**Contribution:** 1
**Rating:** 4
**Confidence:** 4

**Summary:**

This paper introduces CHMQA, a visual question answering task that focuses on free-text answering of consumer health questions. To study this setting, the authors curate HealMQA, a benchmark dataset built from real user questions that annotators rephrase to reference an image, pair with a relevant medical image retrieved from authoritative websites, and answer in clinician-written sentences across 17 topics. The paper benchmarks eight multimodal LLMs under zero- and few-shot prompting using BERTScore, SacreBLEU, ROUGE, and a small human study by two physicians. Results suggest decent textual quality and safety but weaker image-text integration among tested systems.

**Strengths:**

1. The problem itself is interesting, targeting an emerging, high-impact use case: consumer health queries that often mix vague text with a clarifying photo.
2. The dataset design is mostly sound with clear annotation steps, yielding examples that resemble what end users might ask multimodal assistants.
3. The human evaluation, while small, adds a clinically meaningful lens that automated n-gram metrics cannot provide.
4. The paper is well-written and easy to follow.

**Weaknesses:**

1. I am concerned about the pairing between the image and the real scene behind the user's description in the proposed dataset. The image is not user-provided, but retrospectively retrieved to match a textual visual term. This creates **two layers of distributional shift**: (1) the real scene may contain key features that are not described by the users due to their non-professionalism. (2) the image retrieved via Google search may not reflect the user’s description. Such weak alignment makes the HealMQA dataset less suitable for benchmark curation which poses higher standard for validated data than training data curation.

2. Is there an ablation study to quantify the potential image-query misalignment above? For example, start the evaluation with real user image-text pairs and then replace the paired images with ones retrieved from Google and see how the evaluation results will change.

3. Inter-annotator “agreement” via BERTScore on a small pilot is an unusual proxy that conflates style and semantic overlap and can reward superficially similar wording. And I am not sure if 0.7257 should really be seen as “high level similarity and hence, agreement”. This is a potentially large gap, particularly in the medical scenario. A more feasible solution is to use clinically oriented rubric agreement (e.g., structured checklists for differential, red-flag triage, etc) and inter-rater reliability on those rubrics (κ), not just embedding similarity. The current claim of high agreement feels weakly supported.

4. Is there any data leakage detection for the benchmark dataset? Note that all images and texts are directly from the Internet and they might already be consumed by existing MLLMs trained on web-crawled data, making the evaluation results less valuable.

5. Evaluation under-emphasizes visual grounding. The automatic metrics are text-to-text overlap and do not verify whether an answer actually uses image evidence correctly. Although the human study notes “multimodal consistency” as the weakest dimension, again, the experiment is too small (20 items, 2 raters) to draw confident conclusions.

6. I am also concerned about licensing and provenance. “Open access” or preference for .edu/.gov domains is not a license. It is unclear whether images are redistributed, and whether usage complies with original site terms and patient privacy constraints.

7. Some typos. For example, in Line 200, “With HealMQA, we the creation...”. In Line 261, “addresses the user’s.”

**Questions:**

Will this benchmark dataset be open-source?

---

> ### Author Response · Authors · 2025-11-25
> **Author response to Reviewer ws9F**
>
> We thank the reviewer for their comments on our work. We would like to clarify the following points in response to the reviewer’s comments.
>
> 1. **Matching images and query**
>    We have established the direct relationship between the image and query by adapting the following steps in the data creation process:
>    a) Image Retrieval: This is the first step of the dataset creation process. We retrieve the relevant images for the query by extracting the visual terms from the original query.
>    b) Question Rephrasing: The question is then rephrased by substituting the visual terms with neutral references such as “this”, “these”, or “see image”  to transform the original question into the multimodal query accompanied by the appropriate relevant images.
>    c) Answer Formulation: The transformed multimodal query is used to formulate the answer covering both the modality (rephrased textual query and the image) while constructing the answer.
>    d) Answer Consistency to the Image: We also follow a consistency check where, if the annotator is not confident that there is any image that represents the user’s descriptions, the question is skipped and not included in our dataset.
>
> 2. **Evaluating image-query mismatch**
>    We agree that the experiment suggested by the reviewer would result in an interesting and useful analysis. However, the reason to use google images is because of the lack of availability of real user image text pairs for consumer health queries. We are investigating the suggested experiment and will present our results by the end of the discussion period.
> 3. **Inter-annotator agreement**
>    We thank the reviewer for their suggestion of an interesting approach to calculating inter-annotator agreement. We are exploring options for adapting such a questionnaire for our annotators to measure inter-annotator agreement better.
> 4. **Data Leakage**
>    The main contribution of this dataset is i) connecting publicly available user questions to publicly available images, and ii) providing expert-quality answers to the user questions. As the answers are newly written by our expert annotators, there is no concern about leakage of these answers to the LLMs at this point. For the image+text pairs, while both of them are separately available in the public domain, they do not presently exist with this connection between them and we do not believe there is any concerns with leakage of the user questions and images when taken as a single multimodal input.
> 5. **Evaluation of visual grounding**
>    As suggested by another reviewer, we plan to deploy an LLM as a judgment approach for verifying that the answer utilises the image evidence. We also plan to expand the human study to include more samples and evaluators, and will update you once that is conducted.
> 6. **Copyright concerns and Open-source dataset**
>    Our image acquisition process relies on Google image search. A natural drawback of this approach is the use of copyrighted images in the dataset. However, as is common practice \[1, 2\] when working with such types of data, we do not intend to share copies of the data; instead, we will share links to the dataset images. We will host the benchmark locally, where researchers would be able to submit their systems and/or predictions to ensure consistent testing. The answer annotations would be publicly released.
>
> We thank the reviewer for pointing out the mentioned typos and will rectify them in the updated version. We request the reviewer to reconsider their rating of our work with respect to the provided responses to their comments.
>
> \[1\] Piyush Sharma, Nan Ding, Sebastian Goodman, and Radu Soricut. 2018\. Conceptual Captions: A Cleaned, Hypernymed, Image Alt-text Dataset For Automatic Image Captioning. In Proceedings of the 56th Annual Meeting of the Association for Computational Linguistics (Volume 1: Long Papers), pages 2556–2565, Melbourne, Australia. Association for Computational Linguistics.
>
> \[2\] Desai, K., Kaul, G., Aysola, Z.T., & Johnson, J. (2021). RedCaps: web-curated image-text data created by the people, for the people. Thirty-fifth Conference on Neural Information Processing Systems Datasets and Benchmarks Track.

---

> > ### Comment · Reviewer_ws9F · 2025-11-26
> >
> > Since the authors are still busy preparing the results, I will wait until the full response is carried out. No hurry
> >
> > However, there is one point I can now provide the feedback (but I will not ask the author for immediate response).
> >
> > **1. On the pairing between the image and the real scene behind the user's description in the proposed dataset.**
> >
> > The authors explain how the images were paired, but fail to justify why this pairing is clinically valid. A Google-retrieved image of a general symptom (e.g., a 'rash') inevitably differs from the specific, unobserved scene the user actually saw and was describing, particularly given that vague textual descriptions can correspond to multiple distinct pathologies due to individual differences.
> >
> > Consequently, the annotators (even professional), in my opinion, are essentially forced to interpret the retrieved proxy image rather than the unavailable ground truth of the user's actual situation. This risk creating a circular dependency, where **the answer fits the image perfectly only because the answer was derived from that specific image**. This does not guarantee that the system can diagnose the actual disease the user was originally asking about.
> >
> > That's why I think the dataset might not qualify as a benchmark dataset that requires much higher credibility than a training dataset.

---

> > > ### Author Response · Authors · 2025-12-03
> > > **Response to reviewer ws9F**
> > >
> > > We would like to thank the reviewer for their response. We would like to acknowledge a few ways how the question formulation remains valid.
> > >
> > > First, the image that is selected is not a google search of the “general symptom” such as “rash”. The query is the complete visual term which is a lot more descriptive, for example, as shown in figure 2 - “a red rash-type blemish on his cheek”.
> > >
> > > Second, the image retrieved is not the first search result but rather a carefully selected image based on the overall context of the question. For example, as shown in figure 2, outside of the visual term, there are cues in the overall question such as “8 year old son”. The image selected is that of a young boy fitting this overall context. This is in contrast to the first few results on querying the visual term which results in images that are from a clearly different demographic.
> > >
> > > We agree that the selected image may differ from the original observed scene. We do not claim that the multimodal question that is part of HealMQA is a perfect approximation of the real user question. However, we believe that it is true that the image + text pair is a valid approximation of the type of query users present on online forums in terms of query type and language used. We do not claim that we are diagnosing the original textual query of the user, but rather we are diagnosing queries that mimic user queries and the answer is medically validated for these queries. In our opinion, it is still an important test for LLM systems to provide diagnosis for these proxies of consumer queries.

---

### Official Review · Reviewer_erVX · 2025-10-28

**Soundness:** 2
**Presentation:** 2
**Contribution:** 2
**Rating:** 4
**Confidence:** 4

**Summary:**

This paper introduces HealMQA, a clinical expert annotated multimodal question answering dataset designed for consumer healthcare queries. It consists of more than 1k real healthcare questions from online sources along with images retrieved from Google and expert written answers across several healthcare domains, primarily focusing on skin conditions. The authors define a new task, Consumer Healthcare Multimodal Question Answering - CHMQA, to evaluate how AI systems handle multimodal queries in a generalized setting. They benchmark eight LLMs, including GPT-5 and open-source models like Llama and Mistral, under zero- and few-shot settings, using automated metrics (BERTScore, ROUGE, BLEU) and human evaluation by medical professionals. Results show that LLMs produce generally accurate and clear answers but often fail to integrate visual information effectively.

**Strengths:**

- The paper introduces the CHMQA task, a novel and consumer focused multimodal question answering setup that addresses the lack of datasets combining real world consumer facing health queries with visual inputs, bridging the gap between professional medical QA systems and everyday user needs.
- It includes manual annotation and validation by licensed medical experts, providing clinically grounded answers and a structured evaluation of LLM outputs across key medical criteria, though the scale of expert involvement remains relatively limited.

**Weaknesses:**

- The fundamental problem with this benchmark is that there is no relationship between the image and the user query. The quality of the benchmark would simply depend on the quality of the images retrieved by the annotators. Obtaining exact images that accurately matches the user description is quite challenging and is not a scalable approach. This would also lead to unwanted biases in the benchmark where the same image from google could be used for multiple similar user queries.
- The evaluations are not comprehensive enough, Table 2 is missing many state of the art open and close sourced models like – Gemini 2.5 Pro [1], Gemini 2.5 Flash and medical models like Google’s MedGemma[2], Huatuogpt[3], Bimedix2[4] and LLaVA-Med.
- Additionally models like Gemini 2.5 Pro and Gemini 2.5 Flash have muti-image support which helps them take advantage of the few-shot setting. Currently, as mentioned in A.3 models like Llama and mistral are evaluated by giving only the text input which does not incentivize them to use multimodal signals in the few shot setting.
- When evaluating model responses in an open-ended context, the most effective current approach is to employ an LLM as a judge. Alternative methods, such as embedding-based metrics or BERTScore, even those adapted for clinical domains, fail to fully capture the nuanced intricacies of natural language.

[1] *Comanici, Gheorghe, et al. "Gemini 2.5: Pushing the frontier with advanced reasoning, multimodality, long context, and next generation agentic capabilities." arXiv preprint arXiv:2507.06261 (2025)*

[2] *Sellergren, Andrew, et al. "Medgemma technical report." arXiv preprint arXiv:2507.05201 (2025).*

[3] *Chen, Junying, et al. "Huatuogpt-vision, towards injecting medical visual knowledge into multimodal llms at scale." arXiv preprint arXiv:2406.19280 (2024).*

[4] *Mullappilly, Sahal Shaji, et al. "Bimedix2: Bio-medical expert lmm for diverse medical modalities." arXiv preprint arXiv:2412.07769 (2024).*

[5] *Li, Chunyuan, et al. "Llava-med: Training a large language-and-vision assistant for biomedicine in one day." Advances in Neural Information Processing Systems 36 (2023): 28541-28564.*

**Questions:**

Please address the above weaknesses.
- What instructions were given to the annotators for the benchmark creation, please include detailed instructions in appendix to ensure transparency and reproducibility.
- Line 106: “We provide a detailed human analysis of LLM” use correct article “a”.
- Line 202: “With HealMQA, we the creation of a widely useful” please correct the grammar.

---

> ### Author Response · Authors · 2025-11-25
> **Author response to reviewer erVX**
>
> We thank the reviewer for their comments on our work. We would like to clarify the following points in response to the reviewer’s comments.
>
> 1. **Scaling of the dataset**
>    Our primary motivation for the collection of this data was to evaluate multimodal LLMs on this unique category of medical knowledge for which no existing data resources or evaluations exist. We acknowledge the lack of data resources for multimodal consumer healthcare limited our initial collection of the data. We have explored other data sources \- such as the r/AskDocs subreddit which contains user questions along with real user provided images. We plan to include such data sources in our expansion of the dataset by pairing those questions with answers annotated by our medical experts.
>    While scaling is tough, as pointed out by the reviewer, we are committed to invest our resources in expansion of this dataset and are currently expanding the dataset by about double in the coming weeks. At the same time, we believe that this dataset, even at its current scale, is very much needed, as currently thousands of users rely on these LLMs for general medical advice, with no current multimodal datasets to assess the capability of these LLMs in providing said medical advice.
> 2. **Relationship between image and queries**
>    We have established the direct relationship between the image and query by adapting the following steps in the data creation process:
>    a) Image Retrieval: This is the first step of the dataset creation process. We retrieve the relevant images for the query by extracting the visual terms from the original query.
>    b) Question Rephrasing: The question is then rephrased by substituting the visual terms with neutral references such as “this”, “these”, or “see image”  to transform the original question into the multimodal query accompanied by the appropriate relevant images.
>    c) Answer Formulation: The transformed multimodal query is used to formulate the answer covering both the modality (rephrased textual query and the image) while constructing the answer.
>    d) Answer Consistency to the Image: We also follow a consistency check where, if the annotator is not confident that there is any image that represents the user’s descriptions, the question is skipped and not included in our dataset.
>
> 3. **Expansion of evaluations**
>    We have conducted additional experiments and would like to share the results for the models listed and requested by the reviewer here for the zero-shot setup.
>
> | *Model* | *BertScore F1* | *SacreBLEU* | *ROUGE\_1* | *ROUGE\_L* |
> | :---- | ----- | ----- | ----- | ----- |
> | gemini-2.5-flash | 0.33 | 2.95 | 0.26 | 0.16 |
> | BiMediX2 | 0.27 | 2.03 | 0.20 | 0.13 |
> | deepseek-VL2 | 0.24 | 1.75 | 0.18 | 0.12 |
> | LLaVA-Med | 0.26 | 1.96 | 0.20 | 0.13 |
> | MedGemma | 0.31 | 3.40 | 0.25 | 0.17 |
>
>
>
> 4. **Use of LLM as a judge**
>    We thank the reviewer for this constructive suggestion. We are currently running the evaluation using the LLM as a judge setup and will provide an update once we have finished those experiments.
> 5. **Detailed annotation guidelines**
>    We thank the reviewer for this suggestion. We will add the detailed guidelines provided to the annotators to the appendix in the revised version.
>
> We request the reviewer to reconsider their rating of our work with respect to the provided responses to their comments.

---

### Official Review · Reviewer_ngwa · 2025-11-01

**Soundness:** 4
**Presentation:** 3
**Contribution:** 1
**Rating:** 2
**Confidence:** 5

**Summary:**

The paper introduces HealMQA, a benchmark dataset for consumer-healthcare multimodal question answering (CHMQA), featuring around 1k real world user questions paired with medically validated images and expert-written answers. The dataset is positioned as bridging the gap between unimodal medical QA datasets and VQA style tasks. The authors benchmark eight large language models under zero/few‐shot settings.

**Strengths:**

- The topic is timely and socially relevant given the growing use of foundation models for healthcare advice.

- The dataset is carefully annotated by medical professionals and covers diverse consumer health topics, enhancing realism and safety awareness.

- The paper is well written, provides a clear task definition and methodology, and complements automatic metrics with human evaluation.

**Weaknesses:**

- The technical novelty and dataset scale are limited. HealMQA contains only about 1 k examples, which is small compared to recent multimodal medical benchmarks.

- The authors do not sufficiently compare or position HealMQA relative to large and diverse multimodal datasets that already address similar challenges at greater scale and modality coverage. For example:
-- EHRXQA (Bae et al., NeurIPS 2023): integrates textual EHR data and chest X-ray images for multi-modal question answering on structured clinical data.
-- MedTrinity-25M (Xie et al., arXiv 2024): provides 25 M multimodal medical pairs across 10 modalities with multi-granular annotations.
-- WorldMedQA-V (Matos et al., Findings NAACL 2025): a multilingual, multimodal benchmark for medical reasoning and LMM evaluation across diverse clinical domains.

- HealMQA lacks comparison with these stronger baselines and does not clarify its unique contribution beyond the “consumer-health” focus.

- The image acquisition process is insufficiently documented, raising reproducibility and bias concerns.

- The evaluation focuses on generic language metrics that poorly reflect multimodal comprehension, with limited analysis of grounding, hallucination, or safety violations.

- Finally, the dataset’s scale and diversity are inadequate for benchmarking large multimodal models, and the paper does not provide clear evidence of multimodal benefit (i.e., how much images improve text-only performance).

**Questions:**

Add ablations showing how multimodal input (image + text) improves over text-only baselines.

Address copyright, licensing, and bias issues in image sourcing.

Provide a detailed ethical statement covering dataset release, privacy, and intended-use limitations.

---

> ### Author Response · Authors · 2025-11-25
> **Author Response to Reviewer ngwa**
>
> We thank the reviewer for their comments on our work. We would like to clarify the following points in response to the reviewer’s comments.
>
> 1. **Limited technical novelty and dataset scale**
>    We would like to highlight the following novel characteristics of the HealMQA dataset:
>    1. This is the first dataset that focuses on consumer healthcare queries across multiple medical domains including Skin Conditions & Dermatology, Eye & Vision Problems, Dental Health, General Health & Wellness, Hair & Scalp Issues, ENT \- Ear Nose Throat, Infections & Immune System, Infections & Immune System, Cancer & Oncology, Sexual Health, Musculoskeletal Health, Cardiovascular Health, Women's Health, Nutrition & Diet, Pediatric & Geriatric Health, Endocrine & Metabolic Health, Respiratory & Urological Health
>    2. This is a first-of-a-kind dataset with answers annotated by medical experts/doctors for these categories, without relying on LLM generated answers or objective style questions which are automatically answered.
>    3. This is the only benchmark that tests LLMs on **multimodal** consumer queries.
>
>    While we agree that in the broad context of LLM benchmarks, the dataset scale might appear limited, it is primarily due to the high quality of the dataset samples. As part of this dataset, all annotations are conducted by licensed medical professionals who have firsthand experience in answering patient queries in a clinical setting. As a result, the annotation process is expensive and laborious. We undertake this expensive and time-consuming procedure to ensure that LLMs of the future can be validated against high-quality human-annotated data. We are in the process of further expanding the data by around 1500 samples. We will provide updates on this increased dataset size by the end of the discussion period.
> 2. **Inadequate comparison to large-scale medical datasets**
>    We acknowledge the existence of the large-scale datasets pointed out by the reviewer and will update our related works section to introduce these datasets, providing a wider range of comparisons. However, at the same time, we do not believe that our dataset is comparable in task and/or quality to any of the datasets mentioned. Below, we mention some key facts about each of the mentioned datasets and why we believe these are not comparable.
>    1. EHRXQA \- This dataset is about specialised reports such as Chest X-ray which are inaccessible by general consumer. The answers are also objective style answers such as yes/no or entity identification.
>    2. MedTrinity- This dataset is about specialised reports from MRI, Pathology etc which are not accessible by general consumers. The descriptions provided in this dataset are all generated by LLMs with limited validation of these generated descriptions.
>    3. WorldMedQA-V \- This is a small scale dataset with only 726 QA samples from medical exams of 4 countries. This dataset has objective style answers which are not comparable to our free-form text generation. Further, these questions are also related to specialised reports such as x-ray scans as shown in their paper.
>
>
>
> 3. **Consumer-health focus as a contribution**
>    As shown above, all of these datasets have limited applicability in the consumer healthcare setting, which is our primary focus. These datasets are helpful in the identification of conditions of specialised medical reports; however, general patients approaching doctors for an initial diagnosis do not have access to these expensive medical reports. Access to medical facilities for producing specialised scans may also be prohibitively limited in certain parts of the world due to cost and/or accessibility. Our dataset is useful for benchmarking LLMs on the kind of queries that thousands of real users ask LLMs every day, with image data that represents the kind of images they might capture with their mobile cameras.
> 4. **Limited comparison with stronger baselines**
>    In section 5 of the manuscript, we provide details about our experiments benchmarking the dataset. We provide results with some of the strongest-performing multimodal LLMs, such as GPT-5 and o3. These represent some of the strongest LLMs that consumers use on a daily basis. We have also expanded our experimentation to include a wider array of LLMs, including Gemini-2.5-flash, Deepseek VL2, LLaVA-Med, MedGemini and BiMediX2. These results are as follows:
>
>
> | *Model* | *BertScore F1* | *SacreBLEU* | *ROUGE\_1* | *ROUGE\_L* |
> | :---- | ----- | ----- | ----- | ----- |
> | gemini-2.5-flash | 0.33 | 2.95 | 0.26 | 0.16 |
> | BiMediX2 | 0.27 | 2.03 | 0.20 | 0.13 |
> | deepseek-VL2 | 0.24 | 1.75 | 0.18 | 0.12 |
> | LLaVA-Med | 0.26 | 1.96 | 0.20 | 0.13 |
> | MedGemma | 0.31 | 3.40 | 0.25 | 0.17 |

---

> > ### Author Response · Authors · 2025-11-25
> > **Response continued**
> >
> > 5. **More description of Image Acquisition**
> >    Our image acquisition process relies on Google image search. Annotators are asked to identify the visual term in the textual query (Definition of visual term as given in line 236 of the manuscript: a visual term as any word or phrase in a question that conveys observable characteristics of the body or bodily fluids and could be represented by an image. These typically describe color, shape, size, distribution, or location of an object on the skin or body, as well as visible fluids (e.g., blood, discharge).) and then find relevant images using a google image search of the visual term. From the search results, the annotators select the image which best represents the full textual query. This is done based on the similarity of the query visual term such as closeness to the described symptoms, location of the described abnormality and demographic information present in the user query such as gender and age. If there is no image which supports the full textual query, the annotators are asked to skip that question from the annotation. We will add complete guidelines provided to the annotators in the revised version of this paper.
> > 6. **Evaluation and metric selection**
> >    We agree that the automated metrics currently might be poor indicators of language grounding, hallucinations and safety. As a result, we base most of our analysis on our human evaluation. We also plan to extend automated evaluation to LLM-as-a-judge setup as suggested by other reviewers for better evaluation. We will present these results during the discussion period.
> >
> >
> > We request the reviewer to reconsider their rating of our work with respect to the provided responses to their comments.

---

### Author Response · Authors · 2025-11-27
**Results of LLM-as-a-judge evaluation**

Dear Reviewers,
We have conducted the experiments on the LLM as a judge. The judge model in our experiments is the gpt-5-chat model. The instruction prompt provided to the LLM judge follows the same setup as that for the human evaluation, as described in Appendix A.4. The LLM judge is asked to evaluate the same 4 models \- gpt-5-chat, o3, mistral-medium and Llama-4-Scout on the same 5 identified aspects as the human evaluation \- Medical Accuracy, Coverage and Relevance, Multimodal Consistency, Clarity / Understandability, Risk of Harm (Safety). The definitions for each of these are provided to be the same as those for the human evaluators in Appendix A.4. Additionally, the LLM judge, similar to the human evaluators, is asked to holistically compare all the model generations and rank them in order of 1-4 (lower is better), presented as mean rank in the results.  Similar to the human evaluation setup, the model identities are not disclosed to the LLM judge and proxy names  Model A/B/C/D  are used. Furthermore, to mitigate positional bias, the positions A, B, C, and D are randomly assigned for each sample.  The results of this analysis are as follows:

| Model | Medical Accuracy | Coverage and Relevance | Multimodal Consistency | Clarity / Understandability | Risk of Harm (Safety) | Mean Rank |
| :---- | ----- | ----- | ----- | ----- | ----- | ----- |
| gpt-5-chat | 4.44 | 4.18 | 4.63 | 4.90 | 4.86 | 1.76 |
| o3 | 4.40 | 4.12 | 4.51 | 4.76 | 4.66 | 1.72 |
| mistral-medium | 3.56 | 3.37 | 3.66 | 4.49 | 4.31 | 3.35 |
| Llama-4-Scout | 3.63 | 3.58 | 3.80 | 4.51 | 4.26 | 3.18 |

For comparison, below are the results of the human evaluation, originally presented in Section 5.1:

| Model | Medical Accuracy | Coverage and Relevance | Multimodal Consistency | Clarity / Understandability | Risk of Harm (Safety) | Mean Rank |
| :---- | ----- | ----- | ----- | ----- | ----- | ----- |
| gpt-5-chat | 4.60 | 4.45 | 4.28 | 4.43 | 4.43 | 1.60 |
| o3 | 4.23 | 4.10 | 3.65 | 4.10 | 3.98 | 2.50 |
| mistral-medium | 4.05 | 3.73 | 3.70 | 4.35 | 4.03 | 3.05 |
| Llama-4-Scout | 4.05 | 3.95 | 3.80 | 4.15 | 3.90 | 2.85 |

As shown in these tables, the overall trends in the LLM-as-judge evaluation broadly mirror those of the human evaluation. However, a notable discrepancy emerges in how the o3 model is assessed. The LLM judge consistently rates o3 on par with gpt-5-chat, whereas human evaluators show a clear preference for gpt-5-chat. This suggests that human raters are more discriminative in their assessments, leading to more pronounced score gaps across models. In contrast, the LLM-as-judge produces more compressed score distributions, particularly among the top two and bottom two systems.

These findings also highlight that multimodal consistency remains an important open challenge, especially for weaker open-source models. Moreover, the results underscore that open-source LLMs may be more prone to medical accuracy errors, an aspect we did not infer from the results of the human evaluation which presented high scores of medical accuracy even for open-source models. This points to a continued need for developing open-source models that better mitigate such errors.

---

### Meta-Review · Area_Chair_mCgT · 2026-01-08

**Summary:**

This paper introduces HealMQA, a clinically expert-annotated multimodal dataset for benchmarking large language models on consumer healthcare question answering. Reviewers found the problem important and appreciated the involvement of licensed medical professionals and the focus on consumer-facing queries. However, concerns were raised about limited dataset scale, weak technical novelty, questionable image-query alignment, and whether the dataset meets the credibility bar expected of a benchmark. Reviewers also noted insufficient comparisons to existing multimodal medical datasets and limited evaluation of true multimodal grounding.

**Reviewer Concerns:**

Addressed:

- The authors clarified the dataset’s consumer-health focus and distinguished it from existing clinical or report-based datasets.

- Additional model evaluations and LLM-as-a-judge results were provided.

- Image acquisition and annotation guidelines were explained in more detail, and plans for dataset expansion were stated.



Outstanding:

- Concerns about the clinical validity and realism of pairing Google-retrieved images with real user queries

- Dataset scale remains small for benchmarking large multimodal models

- Limited evidence that images provide measurable benefit over text-only inputs

- Evaluation of visual grounding, safety, and multimodal reasoning remains weak

- Concerns about licensing and benchmark credibility

**Reviewer Scores:**

- Reviewer ngwa: Likely unchanged (reject)

- Reviewer erVX: Possibly slightly improved but still borderline

- Reviewer ws9F: Likely unchanged (borderline reject)

---

### Decision · Program_Chairs · 2026-01-26

Reject